# Protocol for tumour-focused dose-escalated adaptive radiotherapy for the radical treatment of bladder cancer in a multicentre phase II randomised controlled trial (RAIDER): radiotherapy planning and delivery guidance

Shaista Hafeez [1,2] Amanda Webster,[3] Vibeke N Hansen,[4] Helen A McNair,[1,2] Karole Warren-Oseni,[5] Emma Patel,[3] Ananya Choudhury,[6,7] Joanne Creswell,[8] Farshad Foroudi,[9] Ann Henry,[10,11] Tomas Kron,[12] Duncan B McLaren,[13] Anita V Mitra,[14] Hugh Mostafid,[15] Daniel Saunders,[16] Elizabeth Miles,[3] Clare Griffin,[17] Rebecca Lewis,[17] Emma Hall,[17] Robert Huddart[1,2]

**Correspondence to**
Dr Shaista Hafeez;
shaista.hafeez@icr.ac.uk

## ABSTRACT

**Introduction** Daily radiotherapy delivered with radiosensitisation offers patients with muscle invasive bladder cancer (MIBC) comparable outcomes to cystectomy with functional organ preservation. Most recurrences following radiotherapy occur within the bladder. Increasing the delivered radiotherapy dose to the tumour may further improve local control. Developments in image-guided radiotherapy have allowed bladder tumour-focused 'plan of the day' radiotherapy delivery. We aim to test within a randomised multicentre phase II trial whether this technique will enable dose escalation with acceptable rates of toxicity.

**Methods and analysis** Patients with T2-T4aN0M0 unifocal MIBC will be randomised (1:1:2) between standard/control whole bladder single plan radiotherapy, standard dose adaptive tumour-focused radiotherapy or dose-escalated adaptive tumour-focused radiotherapy (DART). Adaptive tumour-focused radiotherapy will use a library of three plans (small, medium and large) for treatment. A cone beam CT taken prior to each treatment will be used to visualise the anatomy and inform selection of the most appropriate plan for treatment.

Two radiotherapy fractionation schedules (32f and 20f) are permitted. A minimum of 120 participants will be randomised in each fractionation cohort (to ensure 57 evaluable DART patients per cohort).

A comprehensive radiotherapy quality assurance programme including pretrial and on-trial components is instituted to ensure standardisation of radiotherapy planning and delivery.

The trial has a two-stage non-comparative design. The primary end point of stage I is the proportion of patients meeting predefined normal tissue constraints in the DART group. The primary end point of stage II is late Common Terminology Criteria for Adverse Events grade 3 or worse toxicity aiming to exclude a rate of >20% (80% power and 5% alpha, one sided) in each DART fractionation cohort.

### Strengths and limitations of this study

► Phase II international multicentre randomised controlled study evaluating a novel adaptive radiotherapy technique (strength).
► Treatment allocation favours 75% of participants receiving novel adaptive radiotherapy techniques (strength).
► Detailed guidance and training are provided for the contouring, planning and delivery of this radiotherapy technique to ensure standardisation across participating centres with robust pretrial and on-trial radiotherapy quality assurance programme (strength).
► Primary end point focus is based on determining safety of treatment based on late grade 3 toxicity scoring (strength).
► Non-comparative trial design (limitation).

Secondary end points include locoregional MIBC control, progression-free survival overall survival and patient-reported outcomes.

**Ethics and dissemination** This clinical trial is approved by the London-Surrey Borders Research Ethics Committee (15/LO/0539). The results when available will be disseminated via peer-reviewed scientific journals, conference presentations and submission to regulatory authorities.

**Trial registration number** NCT02447549; Pre-results

## ARTICLE SUMMARY

We present the first international randomised controlled trial protocol evaluating a dose-escalated tumour-focused image-guided adaptive radiotherapy technique. The study

population are patients with unifocal localised muscle invasive bladder cancer. Patients will be randomised (1:1:2) between standard (control) whole bladder single plan radiotherapy (WBRT), or standard dose adaptive tumour-focused radiotherapy (SART) or dose-escalated adaptive tumour-focused radiotherapy (DART). For those randomised to adaptive tumour-focused radiotherapy groups treatment will be delivered using a library of three plans (plan of the day). If successful, the trial will demonstrate feasibly of multicentre implementation of this new radiotherapy technique and inform design of a future phase III trial to establish the optimum organ preserving treatment option for patients with MIBC.

## INTRODUCTION

Radical management of localised muscle invasive bladder cancer (MIBC) involves either radical cystectomy or a course of daily radiotherapy delivered with radiosensitisation over 4–7 weeks.[1–5] Although both have comparable overall survival outcomes in appropriately selected patients, radiotherapy offers opportunity for cancer cure with functional organ preservation.[6]

Most recurrences following radiotherapy occur within the bladder, the majority of which are believed to occur at the original MIBC tumour site, suggesting persistent occult local disease.[7] The modelled dose-response relationship of MIBC to radiotherapy suggests improved local control and overall survival would be expected at higher doses.[8–10]

The ability to safely increase dose beyond the current accepted standard has been restrained by reliable radiotherapy delivery to the bladder. The bladder is a mobile organ which is subject to marked shape and volume change during the course of treatment.[11–13] This bladder motion means historically up to 57% of fractions (f) incur some element of geographical miss even when safety margins of up to 1.5 cm are applied to create the planning target volume (PTV).[14] The expected consequence of improving bladder radiotherapy targeting would be improved tumour control and reduced treatment-related toxicity.

Optimisation of target coverage has been enabled by technology integrated on current generation linear accelerators which allow a three-dimensional (3D) image known as a cone beam CT (CBCT) to be acquired. This is of sufficient contrast to allow soft tissue visualisation. When acquired immediately prior to treatment, it informs positional adjustment to ensure coverage of target with the radiotherapy plan.[15]

A solution enabled by CBCT soft tissue visualisation is 'plan of the day'. Rather than having a single plan available for treatment, a library of plans of varying PTV bladder sizes can be created to cover the range of expected filling and positional variation of the bladder. A plan which best fits the bladder target with least normal tissue irradiation as seen on CBCT immediately prior to treatment is then selected for use each day.[14] In bladder

cancer radiotherapy treatment delivery based on a library of plans has reported benefit in reducing normal tissue irradiation compared with single plan treatment delivery.[16–19] It is yet to be demonstrated whether this approach translates to improved clinical outcomes.

Tumour-focused radiotherapy delivery may offer further opportunity to reduce normal tissue irradiation. Sparing the uninvolved bladder does not appear to compromise local control but randomised controlled studies have failed to demonstrate statistically significant improvement in toxicity.[20 21] Bladder sparing is unlikely to have been optimally achieved in radiotherapy delivery predating CBCT image guidance given the positional uncertainties, the large margins applied and treatment delivery on an empty bladder.

In a single-centre phase I study (NCT01124682), feasibility and safety of tumour-focused dose escalation to 70 Gy delivered using plan of the day has been demonstrated. The RAIDER trial seeks to examine feasibility of this approach in a multicentre setting and to determine the clinical benefit of bladder tumour-focused dose escalation.

Below, we describe the RAIDER trial protocol with particular emphasis on the radiotherapy procedural aspects, including preparatory imaging, treatment planning and delivery with the aim of providing comprehensive description of the radiotherapy implemented for the study.

### Hypothesis

Tumour-focused dose-escalated adaptive radiotherapy using library of three plans can be translated to multiple centres. It will be well tolerated and offer the opportunity to improve local disease control for patients with bladder cancer.

## MATERIALS AND ANALYSIS
### Study design

RAIDER is an international multicentre, multi-arm, two-stage non-blinded phase II randomised controlled trial conducted in accordance with the Research Governance Framework for Health and Social Care and principles of Good Clinical Practice. The trial is registered on the clinicaltrials.gov database (NCT02447549) and is included in the National Institute for Health Research (NIHR) Clinical Research Network portfolio. The final ethics approved version of the RAIDER trial protocol is provided in the supplementary files (online supplemental appendix 1).

Patients will be randomised (1:1:2) between standard (control) WBRT, SART or DART. Treatment allocation is by minimisation with a random element; balancing factors will be centre, neoadjuvant chemotherapy useand concomitant radiosensitising therapy use. Randomisation will take place centrally by the Clinical Trials and Statistics Unit, The Institute of Cancer Research (ICR-CTSU) within a maximum of 10 weeks prior to the planned radiotherapy start date.

Within the UK, there are two commonly used radiotherapy schedules to treat bladder cancer, both supported by the National Institute for Health and Clinical Excellence.[1] Therefore, to accommodate this practice radiotherapy will be delivered daily in either 20f over 4 weeks or 32f over 6.5 weeks in accordance with the participating centre's standard practice. The choice of fractionation will be confirmed by each site before trial commencement and will be used for all patients at that site. The two fractionation cohorts will be analysed separately for the primary end point.

For stage I, the primary end point is the proportion of participants in the DART group meeting the predefined normal tissue radiotherapy dose constraints. The secondary end points of stage I are recruitment rate and the ability of the participating centres to deliver SART and DART treatment as per protocol.

For stage II, the primary end point is grade 3 or greater toxicity occurring 6–18 months following radiotherapy as assessed using Common Terminology Criteria for Adverse Events (CTCAE V.4). The secondary end points of stage II are acute toxicity as measured by CTCAE V.4, patient-reported outcomes as measured by a number of instruments including the Patient-Reported Outcomes version of the Common Terminology Criteria for Adverse Events (PRO-CTCAE), Assessment of Late Effects of RadioTherapy-Bowel, the King's Health Questionnaire, sexual function questions and the 5-level EQ-5D version. Additional secondary end points include health economic-related measures, locoregional MIBC control, progression-free survival and overall survival.

The trial has a number of exploratory secondary end points related to use of adaptive plans including appropriate identification of plan selection, target coverage and dose volume comparison between control (WBRT) and adaptive (SART and DART) planning.

Figure 1 shows the trial schema and overview of follow-up. Table 1 provides summary of the scheduled prerandomisation, on treatment and post-treatment assessments.

## Participants and eligibility

Total target recruitment is set at a minimum of 240 participants with a minimum 120 be recruited to each fractionation cohort (20f or 32f cohort). The final sample size in each fractionation cohort will be determined as that sufficient to accrue 57 DART patients evaluable for the primary end point of late toxicity.

Patients with histological or cytological confirmation of unifocal (T2-T4aN0M0) transitional cell carcinoma of the bladder suitable for radical daily radiotherapy will be approached for inclusion. Eligible patients should be willing to accept assessment with cystoscopy and follow-up schedule as outlined in table 1.

Patients with multifocal invasive disease or history of other malignancy within 2 years of randomisation except for non-melanomatous skin carcinoma, previous non-muscle invasive bladder tumours and low risk prostate cancer (as defined by National Comprehensive Cancer Network, NCCN risk stratification as T1/T2a, Gleason 6 Prostate Specific Antigen (PSA) <10) will be excluded. Those with bilateral prosthetic hip replacements, previous history of radiation to the pelvis or other contraindication to pelvic radiotherapy, for example, inflammatory bowel disease will also be excluded.

## Study treatment

All participants should have had a transurethral resection of the bladder tumour (TURBT) with completion of a bladder map by the performing urologist to aid tumour localisation for radiotherapy. Insertion of fiducial markers to further assist tumour localisation for radiotherapy is also recommended at the time of cystoscopy. Neoadjuvant chemotherapy use prior to randomisation is permitted and encouraged for suitable patients.

Radiotherapy should be planned to commence within a maximum of 10 weeks after randomisation or neoadjuvant chemotherapy completion (if used), to allow sufficient time for treatment planning.

Delivery of radiotherapy with concomitant radiosentiser is permitted. Regimes approved for use within the protocol include mitomycin C and 5-fluorouracil,[2] gemcitabine,[22] cisplatin[23] or carbogen.[3] Each centre should aim to use the same regimen for all their participants. Where this is not possible appropriate substitution is permitted for that participant following discussion with the RAIDER lead investigators.

Participants allocated to the WBRT (control) group will have one radiotherapy plan created treating the whole empty bladder to either 64 Gy in 32f or 55 Gy in 20f. A CBCT scan acquired just prior to treatment delivery can be used by the local investigators to inform an online position correction in accordance with National Radiotherapy Implementation Group Report on Image-Guided Radiotherapy (IGRT)[15] and standard local practice.

Participants allocated to the adaptive tumour-focused planning groups (SART and DART) will have three radiotherapy plans generated a small, medium and large plan. The bladder tumour boost volume will be treated to either standard dose (64 Gy in 32f or 55 Gy in 20f) or escalated dose (70 Gy in 32f or 60 Gy in 20f) depending on whether the participant is allocated to SART or DART, respectively. The uninvolved bladder will receive a lower planned dose either 52 Gy in 32f or 46 Gy in 20f depending on fractionation cohort irrespective of SART or DART randomisation. A CBCT taken immediately prior to each treatment delivery will be used to select the most appropriate 'plan of the day' depending on the bladder volume and shape. A second trained individual verifies the plan selected for treatment.

Plan selection is authorised to be carried out only by radiographers or other delegated practitioners who have attained concordance with the gold standard PTV selection through the Radiotherapy Trials Quality Assurance Group (RTTQA) IGRT credentialing for UK centres and Trans Tasman Radiation Oncology Group (TROG) IGRT

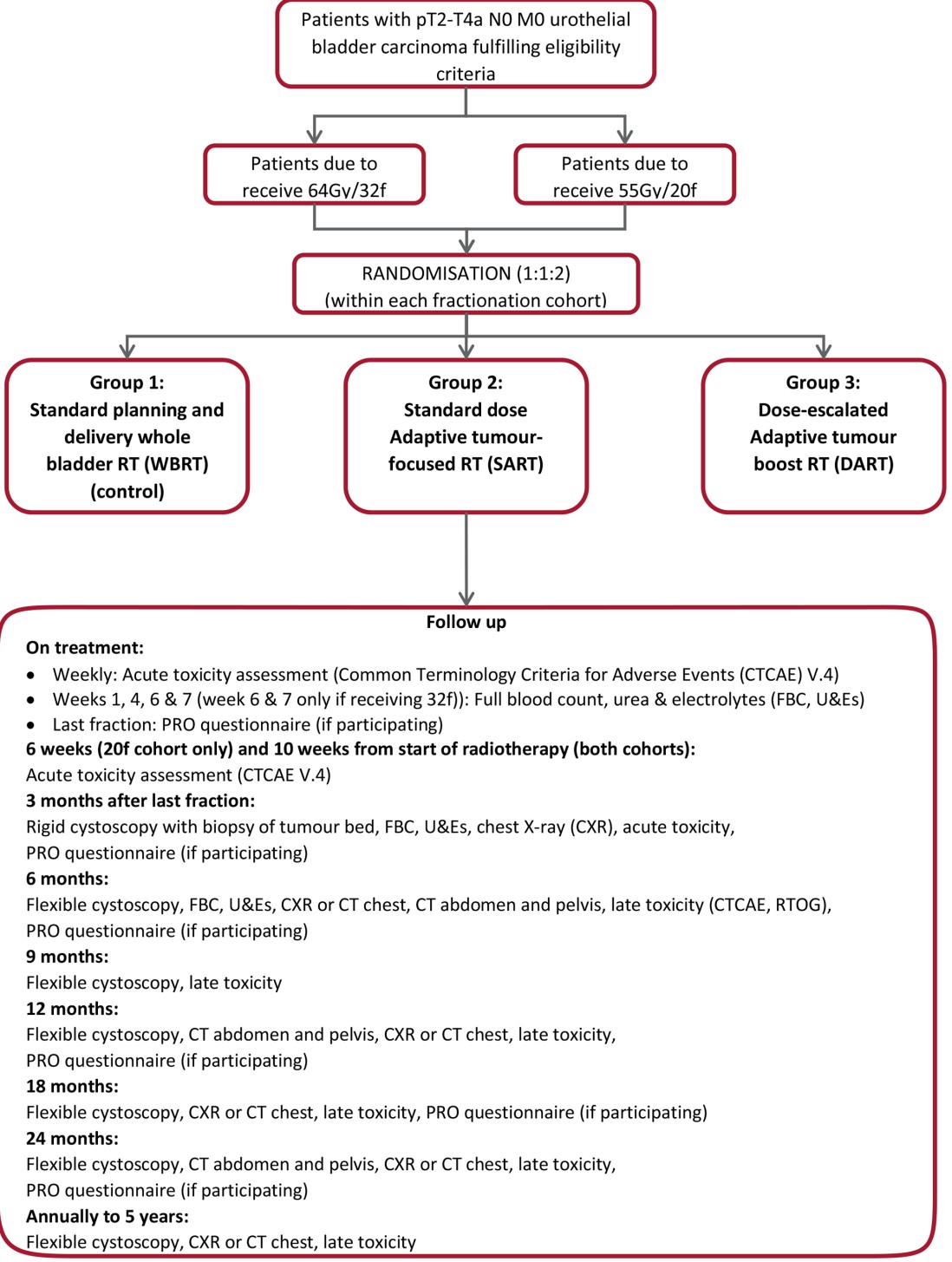

**Figure 1** Trial schema. f, fraction; PRO, patient-reported outcome; RTOG, Radiation Therapy Oncology Group.

credentialing for Australian and New Zealand centres. This is to ensure all those participating in plan selection have the necessary advanced skill level required for the study.

A comprehensive quality assurance (QA) programme has been implemented for the RAIDER trial. This includes pretrial and on-trial components. Selection of appropriate treatment plans for the adaptive planning group will also be independently monitored during patient recruitment as part of the radiotherapy QA process.

**Radiotherapy planning and delivery**
**Radiotherapy planning CT scan**
Bladder preparation procedures vary depending on randomisation group. For WBRT, an empty bladder is required. Patients should be asked to abstain from drinking fluids for 30 min before the scheduled planning CT scan and are required to void their bladder immediately before the planning CT scan is acquired (CT0).

For both SART and DART groups patients are instructed to void their bladder and then drink 350 mL of water. Two

**Table 1** Schedule of assessments

| Visit/Assessment | Preneoadjuvant chemotherapy (if given) | Prerandomisation* | Pre-RT | On treatment | 6 weeks after start RT† | 10 weeks after start RT | 3 months after end RT | 6 months after end RT | 9 months after end RT | 12 months after end RT | 18 months after end RT | 24 months after end RT | Annually to 5 years thereafter | Annually thereafter | At recurrence/disease progression |
|---|---|---|---|---|---|---|---|---|---|---|---|---|---|---|---|
| Radiological assessment‡ | X | X | | | | | | | | | | | | Disease status and survival | Treatment according to local practice, follow-up for disease status and survival |
| TURBT with completion of bladder map | X | X | | | | | | | | | | | | | |
| Placement of fiducial markers (optional) | X | X | X§ | | | | | | | | | | | | |
| Assessment of symptoms/toxicity | | X¶ | X¶ | X¶** | X¶ | X¶ | X¶ | X†† | X†† | X†† | X†† | X†† | X†† | | |
| Full blood count, urea and electrolytes | | | X | X‡‡ | | | X | X | | | | | | | |
| PRO questionnaire (if participating) | | X | X | X§§ | | | X | X¶¶ | | X¶¶ | X¶¶ | X¶¶ | | | |
| Rigid cystoscopy and biopsy of tumour bed | | | | | | | X | | | | | | | | |
| Chest X-ray | | | | | | | X | | | | | | | | |
| Flexible cystoscopy | | | | | | | | X | X | X | X | X | | | |
| CT of abdomen and pelvis | | | | | | | | X | | X | X | X | | | |
| Chest X-ray or CT chest | | | | | | | | X | | X | X | X | X | | |
| Health resource utilisation | | | | X | | | X | X | | X | X | X | | | |

*For patients who have not received neoadjuvant chemotherapy.
†For patients in the 20f cohort only.
‡Recommended imaging: MRI pelvis, CT chest and abdomen. Minimum acceptable is chest, abdomen, pelvis CT or CT chest and CT urogram.
§For patients who received neoadjuvant chemotherapy.
¶CTCAE V.4.
**Weekly on treatment.
††CTCAE V.4 and Radiation Therapy Oncology Group (RTOG).
‡‡During weeks 1, 4 and 6 (week 6 only if receiving 32f).
§§At last fraction.
¶¶Questionnaires administered to UK participants by the Clinical Trials and Statistics Unit, The Institute of Cancer Research from 6 months onwards.
CTCAE, Common Terminology Criteria for Adverse Events; f, fraction; RT, radiotherapy; TURBT, transurethral resection of the bladder tumour.

planning scans are acquired, the first at 30 min following drinking (CT30) and the second 60 min following drinking (CT60). No voiding is permitted between the two scans. However, if voiding is unavoidable because of patient discomfort, then only the available CT30 scan is used for planning.

Given bladder deformation can occur with a loaded rectum, all participants should be encouraged to evacuate their bowels of flatus and faeces prior to acquisition of the radiotherapy planning scanning. The use of microenemas is permitted if it is standard local practice but is not mandated.

All patients will be positioned supine with arms comfortably positioned out of the radiotherapy field using appropriate immobilisation techniques for planning CT scan acquisition. CT slices of ≤3 mm thickness will be obtained from at least 4 cm above the dome of the bladder to 2 cm below the ischial tuberosities. No oral or intravenous contrast is required.

The planning CT scan is exported via DICOM transfer to the radiotherapy treatment planning system for target and organs at risk (OAR) localisation. Bladder filling occurring between CT30 and CT60 scans is determined for those randomised to SART or DART. This is achieved by fusing both CT30 and CT60 data sets and contouring the bladder on both scans. If the difference in bladder volume between the two scans is <50 mL, that is, no significant bladder filling occurs, then all target and OAR contours are created using CT30. If difference in bladder filling is >50 mL, that is, bladder filling occurs, the target volumes for large plan is created using CT60 anatomy.

### Target volume definition

Volumes will be defined according to the International Commission on Radiation Units and Measurements (ICRU) report 50, supplement report ICRU 62, and ICRU 83.[24] Consistent structure naming convention for target volumes and organs at risk is adopted for all patients participating within the trial.

The gross tumour volume (GTV) is defined as the bladder tumour or the resected tumour bed. It is delineated using position of fiducial markers (where available), diagnostic imaging (prior to neoadjuvant chemotherapy where applicable) and the surgical bladder map (where available). When delineating the tumour any extravesical tumour should be included in the GTV. If no tumour is visible then the appropriate section of the bladder should be included based on surgical bladder map following discussion with the urologist who performed the TURBT. Alternatively, repeating the cystoscopy and placing fiducial markers adjacent to resected bladder tumour scar should be considered.

The clinical target volume (CTV) is contoured to encompass the GTV, the whole bladder and any area of extravesical spread. The CTV should also include 1.5 cm of prostatic urethra in male patients or 1 cm of urethra in female patients if tumour is at the base of bladder or if distant carcinoma in situ is present.

A checklist for contouring is provided in the radiotherapy planning and delivery guidelines (online supplemental appendix 2, p. 17). The expansions applied to generate the PTVs are summarised in table 2. The PTV expansion margins were derived from earlier phase I work.[14 16 25]

### Organs at risk delineation

Organs at risk (OARs) are identified as other bowel, rectum and femoral heads in all groups. To quantify normal bladder sparing, the normal bladder outside the boost (PTV2) is also identified for participants in the adaptive tumour-focused radiotherapy groups.

All OARs will be outlined as solid structures by defining their outer wall. The rectum is outlined to include the full circumference and rectal contents. The rectal outlining should extend from the lowest level of the ischial tuberosities to the rectosigmoid junction which identified as the level at which there is an anterior inflection of the bowel, best appreciated on sagittal reconstructions on the CT planning scan.

The small and large bowel (including sigmoid colon) will be outlined as a single structure labelled 'other bowel'. Small and large bowel visible on relevant axial slices of the planning scan will be outlined as individual loops. The cranial extent of 'other bowel' outlining should be 2 cm beyond the superior extent of the standard PTV or large PTV as appropriate.

Both the femoral heads are outlined to the bottom of the femoral head curvature. The femoral necks not included.

The normal bladder outside the boost (PTV2) is created by subtracting the PTV2 from the corresponding CTV.

### Radiotherapy planning

All patients are CT planned. For WBRT, a single plan created using either 3D conformal radiotherapy (3DCRT) with three or four fields, static 5—7-field intensity modulated radiotherapy (IMRT) or volumetric modulated arc radiotherapy (VMAT) technique is permitted. It is accepted that the preferred treatment planning method will vary between participating sites but should be specified in the centre's pretrial process document and be used for all patients enrolled at that centre. Changes in centres preferred planning method from that specified should be brought to the attention of RTTQA.

For participants in the adaptive tumour-focused radiotherapy groups, the planning and dose calculation is done on CT30 data set, therefore all target and OARs volumes are assigned to the CT30 scan. They will have three plans created (small, medium or large) generated from the respective PTV and PTV2 volumes. To enable bladder sparing, these plans are created using either static 5–7-field IMRT or VMAT. The same technique should be used for all patients randomised to adaptive tumour-focused radiotherapy at that centre.

The prescription doses for the PTV are outlined in table 3. All plans should be created with the intention

**Table 2** PTV expansion details

| Patient randomisation | CT data set | PTV | CTV to PTV expansion (cm) | | | | | PTV2 | GTV to PTV2 expansion (cm) | | | | |
|---|---|---|---|---|---|---|---|---|---|---|---|---|---|
| | | | Laterally | Anteriorly | Posteriorly | Superiorly | Inferiorly | | Laterally | Anteriorly | Posteriorly | Superiorly | Inferiorly |
| Group 1 Standard whole bladder (WBRT) | CT0 | PTV | 0.8 | 1.5 | 1.2 | 1.5 | 0.8 | Not applicable | | | | | |
| Group 2 and 3 Adaptive tumour-focused (SART and DART) | CT30 | PTV_Sm | 0.5 | 0.5 | 0.5 | 0.5 | 0.5 | PTV2_Sm | 0.5 | 0.5 | 0.5 | 0.5 | 0.5 |
| | CT30 | PTV_Med | 0.5 | 1.5 | 1.0 | 1.5 | 0.5 | PTV2_Med | 0.5 | 1.5 | 1.0 | 1.5 | 0.5 |
| | If CT60-CT30 bladder filling <50 mL then apply | | | | | | | | | | | | |
| | CT30 | PTV_Lar_30 | 0.8 | 2.0 | 1.2 | 2.5 | 0.8 | PTV2_Lar_30 | 0.8 | 2.0 | 1.2 | 2.5 | 0.8 |
| | If CT60-CT30 bladder filling >50 mL then apply | | | | | | | | | | | | |
| | CT60 | PTV_Lar_60 | 0.5 | 1.5 | 1.0 | 1.5 | 0.5 | PTV2_Lar_60 | 0.5 | 1.5 | 1.0 | 1.5 | 0.5 |

CTV, clinical target volume; PTV, planning target volume.

of achieving the target volume objectives as outlined in table 4. Dose to OARs should be as low possible. The OARs dose volume constraints for both fractionations are summarised in table 5.

The other bowel, rectum and femoral heads constraints for the 32f schedule were derived from previous phase III prostate (CHHiP, convential or hypofractionated high dose intensity modulated radiotherapy for prostate cancer; ISRCTN97182923) and bladder (BC2001; ISRCTN68324339) studies[2 26 27] and from phase I work.[28] The absence of previously defined OARs constraints when dose escalating in 20f meant that the OARs constraints at higher doses were marginally more conservative than if otherwise converted exactly from 32f constraint level using the linear quadratic model alone.[29] The constraints used for the 20f schedule were estimated from the 32f constraint level using the linear quadratic equation assuming that all $\alpha/\beta$ of organs at risk is 3 but the dose constraint is reached in 3 Gy per fraction.

Dose objectives to the PTV should not be compromised to achieve dose to OAR constraints. The recommended hierarchy of planning priorities is providing radiotherapy planning and delivery guidelines (online supplemental appendix 2, p. 27).

For patients randomised to WBRT, it is at the local principal investigator's (PI) discretion to accept the OAR doses. For those randomised to adaptive tumour-focused radiotherapy groups it is recommended that the predefined optimal dose constraints are met for the small plan, and the mandatory constraints for the medium plan wherever possible. It is accepted that the rectum and bowel dose constraints of the large plan may not be met despite adequate optimisation. Assessment of 'other bowel' dose on the large plan represents an overestimation of actual dose compared with 'other bowel' when this plan is actually used to deliver treatment. This is because when the large plan is selected for treatment, a proportion of bowel moves out of the field with bladder filling.

For patients allocated to DART, if the mandatory constraints are not met on the medium plan advice must be sought from the RTTQA team. Decision will be then made by the RAIDER trial team regarding the appropriateness of proceeding at the DART prescription dose or to lowering the prescribed dose as per SART randomisation. It is therefore recommended that the medium plan be optimised first. If patients are not able to receive DART (in either fractionation cohort) for any reason then details of the deviation from allocated treatment will be requested

### Preradiotherapy checks

To minimise risk of error at the time of plan importing, exporting and plan selection, it is recommended that each plan, beam name and ID reflect the assigned plan, for example, Sm_Plan used for labelling the beams making up the small plan in the adaptive tumour-focused radiotherapy groups. It is also important to ensure that the local record and verify systems for

**Table 3** Prescription doses

| Patient randomisation | Volume | 32 fraction cohort | | 20 fraction cohort | |
|---|---|---|---|---|---|
| | | Dose (Gy) | Dose per fraction (Gy) | Dose (Gy) | Dose per fraction (Gy) |
| Group 1 WBRT | PTV_Std | 64 | 2 | 55 | 2.75 |
| Group 2 SART | PTV2 | 64 | 2 | 55 | 2.75 |
| | | 52 | 1.625 | 46 | 2.3 |
| Group 3 DART | PTV2 | 70 | 2.1875 | 60 | 3 |
| | | 52 | 1.625 | 46 | 2.3 |

DART, dose-escalated adaptive tumour-focused radiotherapy; PTV, planning target volume; SART, standard dose adaptive tumour-focused radiotherapy; WBRT, whole bladder single plan radiotherapy.

3DCRT and IMRT cannot mix beams from different plans at the time of exporting or deliver more than one plan at treatment.

### Treatment scheduling

Radiotherapy can start on any day of the week and should be delivered 5 days a week until completion. Interruptions during radiotherapy should be avoided as they have detrimental effect on outcome.[30] All missed fractions are to be reported to the ICR-CTSU and RTTQA team.

In the event of missed fractions due to machine breakdown, bank holiday or any other logistical reason compensation for the missed fraction is advised. This is expected to be achieved by either treating at a weekend or by hyperfractionating, that is, undertaking two fractions a day (ideally on a Friday) with a minimum 6-hour gap between treatments. Should a treatment break occur due to toxicity, centres are advised to contact ICR-CTSU and/or RTTQA. Compensation is not expected in circumstances where missed treatment is a result of radiotherapy-related toxicity.

For those allocated to adaptive tumour-focused radiotherapy groups if plan selection capabilities are unavailable, either because of absence of trained staff, machine breakdown and/or gap day treatment, patients may be treated for up to 5 days using the PTV medium plan without plan selection. These pretreatment CBCTs (if acquired) should be sent to RTTQA for review.

### Treatment delivery

The same patient preparation instructions used at planning CT should be implemented prior to each fraction delivered.

For those patients allocated to SART or DART, CBCT of the pelvis should be acquired prior to each fraction. For those patients randomised to WBRT, pretreatment CBCT should be used in accordance with guidance provided in the NRIG IGRT report.[15] It is therefore expected that this CBCT will inform appropriate corrections (either manual or automatic) to be applied prior to the delivered fraction in accordance with the centre's local practice to ensure that treatment is accurately directed. Any changes made on the basis of the scan including exposures that do not lead to treatment because of patient factors should be reported in the case report forms (CRF) and to RTTQA.

For those randomised to adaptive tumour-focused radiotherapy groups, the pretreatment CBCT is acquired and registered to bone according to the guidance provided in the NRIG IGRT report.[15] An appropriately trained radiographer or practitioner reviews the bone-matched CBCT assessing the bladder size and position in relation to the PTVs and the coverage they provide.

To assist trained radiographers or practitioners with optimal plan selection the following sequential assessment is advised:

i. Following CBCT acquisition, the bladder filling and shape is first checked against CTV_30 contour. If the

**Table 4** Target volume dose objectives

| Volume | Dose constraints | Optimal | Mandatory |
|---|---|---|---|
| PTV2 | $D_{98\%}$ | ≥95% of prescribed dose | ≥90% of prescribed dose |
| | $D_{50\%}$* | – | ±1% of prescribed dose |
| | $D_{2\%}$ | ≤105% of prescribed dose | ≤107% of prescribed dose |
| PTV (PTV–PTV2) | $D_{98\%}$ | ≥95% of prescribed dose | ≥90% of prescribed dose |

*Please note that $D_{50\%}$ constraint refers only to PTV2. PTV $D_{50\%}$ is likely to be exceeded depending on size of PTV2. Therefore, no compromise to PTV2 coverage should be made at the expense of achieving $D_{50\%}$ PTV constraint.
PTV, planning target volume.

**Table 5** Organ at risk dose constraint guide

| Normal tissue | 32 fraction cohort | | | 20 fraction cohort | | |
|---|---|---|---|---|---|---|
| | Constraint | Optimal | Mandatory | Constraint | Optimal | Mandatory |
| Rectum | V30Gy | | 80% | V25Gy | | 80% |
| | V50Gy | | 60% | V41.7Gy | | 60% |
| | V60Gy | | 50% | V50Gy | | 50% |
| | V65Gy | | 30% | V54.2Gy | | 30% |
| | V70Gy | | 15% | V58.3Gy | | 15% |
| Femoral heads | V50Gy | | 50% | V41.7Gy | | 50% |
| Other bowel | V45Gy | 116cc | 139cc | V37.5Gy | 116cc | 139cc |
| | V50Gy | 104cc | 127cc | V41.7Gy | 104cc | 127cc |
| | V55Gy | 91cc | 115cc | V45.8Gy | 91cc | 115cc |
| | V60Gy | 73cc | 98cc | V50Gy | 73cc | 98cc |
| | V65Gy | 23cc | 40cc | V54.2Gy | 23cc | 40cc |
| | V70Gy | 0cc | 10cc | V58.3Gy | 0cc | 10cc |
| | V74Gy | 0cc | 0cc | V61.7Gy | 0cc | 0cc |
| Whole bladder constraint (ie, CTV)* | V60Gy V65Gy | 50% 40% only in DART Otherwise 0% in SART | 80% 50% only in DART Otherwise 5% SART | V50Gy V54.2Gy | 50% 40% only in DART Otherwise 0% in SART | 80% 50% only in DART Otherwise 5% SART |
| Body-PTV (normal tissue) | $D_{1cc}$ | ≤105% of prescribed dose | ≤110% of prescribed dose | $D_{1cc}$ | ≤105% of prescribed dose | ≤110% of prescribed dose |

*Whole bladder (CTV) constraint specified should be used to inform plan optimisation. Bladder outside PTV2 (ie, CTV-PTV2) meeting these contraints will also be collected for reporting of the primary end point.
CTV, clinical target volume; DART, dose-escalated adaptive tumour-focused radiotherapy; PTV, planning target volume; SART, standard dose adaptive tumour-focused radiotherapy.

bladder is of similar size and shape to the CTV at planning (ie, CTV_30), then the small plan should be considered in the first instance for treatment.

ii. The appropriate plan provides suitable coverage of the CTV and boost region by the corresponding PTV and PTV2 contours with minimal normal tissue irradiation.

iii. Manual (soft tissue) moves should be made to ensure the bladder (CTV) is adequately covered while selecting the smallest plan possible to spare normal tissue.

iv. Care should be taken when applying any soft-tissue shifts >1 cm as it can impact on the accuracy of the expected dosimetry. If shifts over 1 cm occur, they should be discussed with the planning department and RTTQA should be contacted following treatment.

v. Manual moves should be undertaken if further optimisation of PTV2 coverage can be achieved. Manual moves prioritising coverage to the boost region over the normal bladder wall is permitted if it avoids excessive normal tissue irradiation that would have occurred by selecting a larger plan.

vi. Finally, the OARs as seen on the CBCT is reviewed and compared with the position on the planning CT. The position of OARs relative to the boost is assessed to ensure that excessive normal tissue does not sit within the PTV2, especially for DART patients. If this is the case, manual move is permitted to minimise normal tissue irradiation but should not be at the expense of target coverage.

vii. A second accredited radiographer or practitioner must confirm selected plan and any additional actions taken. Once agreement has been reached, any necessary couch correction is performed prior to treatment delivery with the selected plan.

Fractions must not be omitted or missed due to unfavourable positioning of normal anatomy such as rectal distention due to flatus or faeces. Additional guidance and potential solutions are provided for scenarios that may arise on treatment are given in the radiotherapy planning and delivery guidelines (online supplemental appendix 2, p. 45). The flow chart of potential interventions is derived from phase I experience previously published.[28]

For example, if the bladder is significantly smaller than the CTV_30 contour at planning, it is likely that the PTV2 boost will be in the incorrect position and, or does not achieve adequate normal bladder sparing. In these circumstances, patients should be removed from the treatment couch, and encouraged to fill the bladder by drinking further, and or increasing the time interval of image acquisition.

In the event that the bladder has overfilled and none of the PTVs provides adequate coverage despite manual moves, the patient should be asked to minimally void

and the CBCT is repeated. If this is not possible, patient should void completely and restart drinking protocol with a reduction in the time interval for CBCT acquisition. In these circumstances, a member of the clinical team should also be notified to ensure the patient is not in urinary retention.

When amending the drinking protocol to optimise patient's anatomy to fit the existing PTV contours, it is advised that one aspect is changed at a time, that is, interval for CBCT acquisition timing or the amount of water that is drunk. This is so the impact of the intervention can be determined and altered for subsequent fractions as required.

If no PTV contours are suitable to cover the target because of rectal gas, then the patient should be removed from the bed and ask to void. The CBCT image acquisition is then repeated. If the PTV contours still are not optimal, it is recommended that the most suitable plan is selected which optimises coverage of PTV2 and minimises the inclusion of OARs is chosen for treatment. If this occurs repeatedly (eg, more than twice in five fractions) RTTQA should be contacted for advice.

All CBCT exposures including those not resulting in treatment should be recorded on the CRF and plan selection form.

In all randomised groups, a post-treatment CBCT should be taken during the first week and once a week thereafter. This CBCT should be reviewed locally to ensure intrafraction filling has been accommodated for at the time of plan selection.

### Radiotherapy protocol compliance programme

The RAIDER trial is subject to radiotherapy QA programme that aims to standardise contouring, planning and delivery of image-guided and adaptive bladder radiotherapy in participating centres. The RTTQA group coordinates the UK QA programme for the study. For Australian and New Zealand participants, this is coordinated by the TROG QA Team.

The QA programme has a pretrial and on-trial component. Each centre will be required to complete the pretrial QA prior to commencing recruitment.

Prior to trial entry, participating centres will be asked to complete a facility questionnaire in order to gauge current local IGRT experience. A separate process document is used to collect task details of all aspects of a complete patient pathway.

The PI at each participating site is asked to contour two benchmark clinical cases as per protocol. One case includes tumour bed GTV as defined by placement of fiducial markers (radio-opaque contrast agent, lipiodol). UK PIs who completed outlining benchmark cases for the preceding phase II adaptive bladder radiotherapy trial (HYBRID Trial, NCT01810757) will be asked to contour only the target volumes as the OARs contouring is unchanged for the RAIDER protocol.[31] Structured feedback to the PI will be provided via RTTQA team.

All participating trial centres will also be required to complete a planning benchmark case. Centres will be provided with access to CT DICOM data and preoutlined structure set. They will be requested to the plan this patient in their own treatment planning system as if randomised to the DART arm. It is the responsibility of the local investigator to ensure that appropriate plan checking QA process is in place at their local institution. Once the three plans of the benchmark case have been created, reviewed and accepted by the local PI, the DICOM CT, dose cubes, RTplan and structure sets are returned to the RTTQA team and structured feedback is provided.

It is a pretrial requirement that all participating centres have both an established IGRT training programme in place for their radiographers and be using CBCT to assess bladder treatment delivery. Trial-specific bladder IGRT competency will be completed through an online plan selection training package, and practical workshop.[32]

The online plan selection training consists of two practice cases each with six CBCTs to work through. Step-by-step instructions with correct plan selections is provided. Following this, a credentialing assessment consisting of 12 plan selections will be carried out. The plan selections and matched reviews will be assessed by RTTQA and structured feedback provided. Only those who meet minimum threshold of concordance of plan selection as predefined by the trial team will be approved for performing RAIDER plan selection. Those who were accredited for plan selection in the HYBRID study[31] or in the TROG 10.1 BOLART trial training (NCT01142102)[33] will not be asked to repeat this assessment.

As part of the on-trial QA, the contouring and planning of at least the first adaptive patient and the first DART will be subject to prospective review by the RTTQA group.

All planning data and treatment delivery data including paired weekly pretreatment and post-treatment CBCTs, registration objects and treatment forms will be collected and reviewed retrospectively by the RTTQA group to ensure adherence to the RAIDER planning and delivery protocol is maintained. Remote retrospective plan selection review will take place for adaptive radiotherapy patients during the trial.

### Statistical considerations

The primary aim of the study is to evaluate the feasibility (stage I) and safety (stage II) of DART. Control (WBRT) and SART treatment groups are included to enable SART to be carried forward to stage II if dose constraints cannot be met in the DART group and to assess equipoise and feasibility of recruitment for any subsequent phase III trial. Prospectively collected contemporaneous toxicity data for WBRT and SART will also allow benchmarking of DART results. Patients are randomised 1:1:2 to maximise information on DART. Recruitment to stage II will continue seamlessly while stage I is evaluated, unless advised otherwise by the Independent Data Monitoring

Committee (IDMC). Patients recruited in stage I will contribute to analysis of stage II.

The sample size of stage I is based on proportion of patients allocated to DART meeting the predefined dose constraints of bladder, bowel and rectum on the medium plan. A patient in the 32f cohort will be defined as meeting the dose constraints if all mandatory constraints of the following are met for the medium plan: rectum constraints at 50 Gy, 60 Gy, 65 Gy and 70 Gy; bladder outside PTV2 at 60 Gy and 65 Gy and small bowel at V55, V60, V65, V70 and V74. A patient in the 20f cohort will be defined as meeting the dose constraints if all mandatory constraints of the following are met for the medium plan: rectum constraints at 41.7 Gy, 50 Gy, 54.2 Gy and 58.3 Gy; bladder outside PTV2 at 50 Gy and 54.2 Gy and bowel at V45.8, V50, V54.2, V58.3 and V61.7.

It is expected that in 80% of DART patients the predefined dose constraints of the medium plan to the normal bladder, bowel and rectum will be met. If this proportion is <50%, it will be concluded that DART delivery is not feasible. Using an A'Hern single stage design (p0=0.5, p1=0.8, 5% alpha and 80% power), 18 patients are required in each DART fractionation cohort. If at least 13/18 meet dose constraints, it will be concluded that DART treatment is feasible; if dose constraints are not met for six or more patients in either fractionation cohort, the IDMC will advise on continuation of the trial with the option of dropping the DART arm in one or both fractionation cohorts and continuing to stage II with randomisation to WBRT versus SART. Stage I will therefore require a total of 72 patients (36 in each fractionation cohort) randomised 1:1:2 between WBRT, SART and DART.

There are no formal early stopping rules for acute toxicity or efficacy but if after six patients have been treated per fractionation cohort, >50% of patients experience acute ≥grade 3 treatment-related toxicity, the IDMC would be asked to advise on suitability of continuation.

Stage II has a non-comparative design aiming to rule out an upper limit of any late ≥grade 3 CTCAE toxicity in each DART fractionation cohort. To be considered evaluable for the primary end point of late toxicity, a patient must receive at least one fraction of allocated treatment and have at least one toxicity assessment performed between 6 and 18 months after completing radiotherapy. It is expected that the proportion of patients in the control group reporting ≥grade 3 toxicity CTCAE toxicity between 6 and 18 months postradiotherapy will be 8%.[20] Again using an A'Hern single stage design (p0 (toxicity free)=0.80, p1=0.92, 5% alpha and 80% power), 57 patients in each DART fractionation cohort will allow a >20% ≥grade 3 toxicity CTCAE toxicity to be excluded. If more than >6/57 evaluable DART patients experience ≥grade 3 toxicity in either fractionation cohort, then the late toxicity threshold will be exceeded and on the IDMC's recommendation the trial could either be stopped or the DART arm dropped.

Allowing for 5% non-evaluability for late toxicity by 18 months gives a sample size of 120 patients (30 WBRT, 30 SART, 60 DART) for each fractionation cohort, that is, a total target sample size of 240. The non-evaluability rate will be monitored and, with IDMC endorsement, cohort recruitment will continue until there are 57 evaluable DART patients per cohort. During stage II, following IDMC review, consideration would be given to dropping the WBRT or SART arms, if it was felt sufficient data had accrued for these arms and it would expedite meeting the aims of the trial. The IDMC will also monitor recurrence rates. If an absolute excess of locoregional recurrence is seen, early termination of the trial would be considered.

For stage I, the primary end point will be presented as the frequency and percentage of randomised patients able to meet the trial dose constraints in the DART group. For stage II, the primary end point will be based on the evaluable population. The proportion of patients with any ≥grade 3 CTCAE toxicity occurring within 6–18 months postradiotherapy will be presented for each randomised treatment group together with the 90% one-sided binomial CI (the 90% two-sided CI will also be presented). A sensitivity analysis will be conducted using a per-protocol population. The per-protocol population will include evaluable patients who received their complete fractionation schedule (either 32f or 20f) according to their randomised allocation group.

The local control rate at 2 years will be presented by treatment group with a 95% CI. Acute and late toxicity will be summarised by frequencies and proportions at each time point by treatment group. Kaplan-Meier methods will also be used to analyse time to local disease progression and overall survival with data presented by randomised group.

## ETHICS AND DISSEMINATION
The trial is approved by the London-Surrey Borders Research Ethics Committee (15/LO/0539).

The first participant was enrolled in October 2015. The study recruitment is scheduled to complete in Spring 2020. It is expected that the trial will report in 2022, following which the results will be disseminated via peer-reviewed scientific journals, conference presentations and submission to regulatory authorities.

### Safety reporting
Data are collected at each trial visit regarding any adverse events graded according to CTCAE V.4 criteria on the CRF. The highest grade observed since the last visit should be reported. All serious adverse events (SAEs) occurring from the start of radiotherapy up to 30 days following the last fraction and any radiotherapy-related ≥grade 3 events occurring between 6 and 18 months are reported to the ICR-CTSU within 24 hours of the PI becoming aware of the event. SAEs should be followed up until clinical recovery is complete or until the condition has stabilised. Any safety concerns will be reported to the main research

and ethics committee by ICR-CTSU as part of the annual progress report.

## Trial monitoring and oversight

A Trial Management Group (TMG) will be set up and will include the Chief Investigator, ICR-CTSU Methodology Lead, co-investigators, identified collaborators, the trial statistician, trial manager and patient representative.

The ICR-CTSU Urology Radiotherapy Trials Steering Committee (TSC) includes a chairperson not directly involved in the trial, and at least two other independent members who will oversee the RAIDER trial. The TSC will meet annually.

An IDMC will be set up to monitor the progress of the trial and will include at least three independent members, one of whom will be a medical statistician. The Committee's terms of reference, roles and responsibilities will be defined in a charter issued by ICR-CTSU. The IDMC will meet in confidence at regular intervals, and at least annually. A summary of findings and any recommendations will be produced following each meeting. This summary will be submitted to the TMG and TSC, and if required, the main REC.

## Patient and public involvement

The RAIDER trial has been reviewed and endorsed by patient and carer representatives from the National Cancer Research Institute (NCRI) Consumer Liaison Group and the NCRI Clinical and Translational Radiotherapy Research Group (CTRad) working group. The CTRad consumer group also approved the proposal for randomisation ratio to be weighted towards participants receiving advanced radiotherapy techniques.

Patient and public involvement began at the protocol design and development stage via national and local consumer oversight committee review. This included the NIHR Biomedical Research Centre radiotherapy studies consumer panel at The Institute of Cancer Research and The Royal Marsden NHS Foundation Trust, and the NCRI Bladder Clinical Studies Group, which includes consumer representation.

Patients who had participated in the phase I bladder radiotherapy studies[25][28] were asked to assess if the burden of involvement required for participation was appropriate. This included review of the patient-reported outcomes questionnaires.

The trial patient information sheet and consent form were reviewed by the South West London Cancer Research Network consumer group. Their feedback was adopted and incorporated into the final version of both documents. Copy of the ethics approved final version of the patient information sheet and consent form are provided in the online supplemental appendix 3.

Patient representation on the TMG advises on day-to-day management of the trial including patient recruitment, and it is expected that they will also participate in dissemination of results via bladder cancer patient groups.

## CONCLUSIONS

RAIDER represents the first randomised trial of dose-escalated adaptive tumour-focused 'plan of the day' radiotherapy and provides a framework for multicentre implementation of this technique. It seeks to investigate whether this approach will allow an increase of radiation dose to be delivered to the tumour with acceptable toxicity. Results will inform the design of a future phase III trial to establish the optimum organ preserving treatment option for patients with MIBC.

**Author affiliations**
[1]Radiotherapy and Imaging, The Institute of Cancer Research, London, UK
[2]Radiotherapy Department, The Royal Marsden NHS Foundation Trust, London, UK
[3]National Radiotherapy Trials Quality Assurance Group (RTTQA), Mount Vernon Hospital, Northwood, UK
[4]Laboratory of Radiation Physics, Odense University Hospital, Odense, Denmark
[5]Joint Department of Physics, The Institute of Cancer Research and The Royal Marsden NHS Foundation Trust, London, UK
[6]Division of Cancer Studies, The University of Manchester, Manchester, UK
[7]Department of Clinical Oncology, Christie NHS Foundation Trust, Manchester, UK
[8]Department of Urology, James Cook University Hospital, Middlesbrough, UK
[9]Department of Radiation Oncology, Austin Health, Heidelberg, Victoria, Australia
[10]Leeds Institute of Medical Research, University of Leeds, Leeds, UK
[11]Department of Clinical Oncology, Leeds Teaching Hospitals NHS Trust, Leeds, UK
[12]Department of Physical Sciences, Peter MacCallum Cancer Centre, Melbourne, Victoria, Australia
[13]Edinburgh Cancer Centre, Western General Hospital, Edinburgh, UK
[14]Cancer Services, University College London Hospitals NHS Foundation Trust, London, UK
[15]The Stokes Centre for Urology, Royal Surrey Hospital NHS Foundation Trust, Guildford, UK
[16]Department of Clinical Oncology, Nottingham University Hospitals NHS Trust, Nottingham, UK
[17]Clinical Trials and Statistics Unit, The Institute of Cancer Research, London, UK

**Acknowledgements** SH, KW-O, HMcN, RL, EH and RH acknowledge this study represents independent research supported by the National Institute for Health Research (NIHR) Biomedical Research Centre at The Royal Marsden NHS Foundation Trust and The Institute of Cancer Research, London. AC is supported by the NIHR Manchester Biomedical Research Centre. The trial was supported by the National Radiotherapy Trials Quality Assurance Team (RTTQA).

**Contributors** All authors met at least one of the criteria recommended by the ICMJE. RH and EH conceived the study design. SH wrote the first draft of the radiotherapy protocol and manuscript. All authors were involved in protocol development and contributed to subsequent revisions of the protocol and manuscript.

**Funding** The RAIDER trial is funded by Cancer Research UK (CRUK/14/016) with programme grants to support the work of the Clinical Trials and Statistics Unit at The Institute of Cancer Research (ICR-CTSU) (grant number C1491/A15955).

**Disclaimer** The views expressed are those of the author(s) and not necessarily those of the NIHR or the Department of Health and Social Care.

**Competing interests** SH reports non-financial support from Elekta (Elekta AB, Stockholm, Sweden), non-financial support from Merck Sharp & Dohme (MSD), personal fees and non-financial support from Roche outside the submitted work; AC reports grants from National Institute of Health Research Manchester Biomedical Research Centre, grants from Cancer Research, UK, grants from Medical Research Council, UK, grants from Prostate Cancer, UK, grants from Bayer, UK, personal fees from Janssen Pharmaceutical, non-financial support from ASCO, grants and non-financial support from Elekta AB, outside the submitted work; AH reports grants

from CRUK, grants from MRC, grants from NIHR, outside the submitted work; TK reports reports grants from Cancer Australia, during the conduct of the study; and his group has a research collaborative agreement with Varian Medical System not related to this project; EH reports grants from Cancer Research UK during the conduct of the study; grants from Accuray Inc., grants from Varian Medical Systems Inc., outside the submitted work; RH reports non-financial support from Janssen, grants and personal fees from MSD, personal fees from Bristol Myers Squibb, grants from Cancer Research UK, other from Nektar Therapeutics, personal fees and non-financial support from Roche outside the submitted work.

**Patient consent for publication**  Not required.

**Provenance and peer review**  Not commissioned; externally peer reviewed.

**ORCID iD**
Shaista Hafeez http://orcid.org/0000-0002-2057-0946

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
