## [Reviewer comments · BMJ Open]

ARTICLE DETAILS

TITLE (PROVISIONAL)	Protocol for tumour focused dose escalated adaptive radiotherapy for the radical treatment of bladder cancer in a multi-centre phase II randomised controlled trial (RAIDER): radiotherapy planning and delivery guidance
AUTHORS	Hafeez, Shaista; Webster, Amanda; Hansen, Vibeke; McNair, Helen; Warren-oseni, Karole; Patel, Emma; Choudhury, A; creswell, Joanne; Foroudi, Farshad; Henry, Ann; Kron, Tomas; McLaren, Duncan; Mitra, Anita; Mostafid, Hugh; Saunders, Daniel; Miles, Elizabeth; Griffin, Clare; Lewis, Rebecca; Hall, Emma; Huddart, Robert

VERSION 1 – REVIEW

REVIEWER	mccm Hulshof Amsterdamumc
REVIEW RETURNED	22-Jul-2020

GENERAL COMMENTS	The manuscript is a protocol of a running randomized phase II study in muscle invasive bladder cancer comparing whole bladder radiation (standard arm), tumor focused radiotherapy and dose escalated adaptive tumor focused radiotherapy. The study will answer a relevant question in bladder sparing treatment and might be able to improve technique and reducing toxicity in BST. The study protocol is too extended and complicated to be described when publishing the results and thus deserves publication in a separate manuscript. As a reviewer of the manuscript it is not my task to comment on the study design itself. My only comment to the authors would be to add under "strength and limitations" the large accepted heterogeneity in treatment and techniques allowed (dose fractionation, chemotherapy schedules and types of drugs, use of fiducials etc) which might bias the toxicity results.
--

REVIEWER	Liam Mannion City, University of London + King's college London. UK
REVIEW RETURNED	22-Jul-2020

GENERAL COMMENTS	This is a protocol for a phase II multi-centre randomised controlled trial on muscle invasive bladder cancer. The protocol is well written and thought out, providing sufficient methodological detail. The only limitation is acknowledged by the authors, in that a comparative analysis of treatment regimens is not possible due to randomisation.
--

REVIEWER	Takahiro Osawa Hokkaido University Hospital, Japan
REVIEW RETURNED	30-Jul-2020

GENERAL COMMENTS	The authors submitted a study in which they tried to test within a randomized multi-center phase II trial whether this technique will enable dose escalation with acceptable rates of toxicity. I think this article is interesting and insightful, and particularly well written. However, the major points you should consider are the following: Major 1. Dose calculation is done on CT30 data set for adaptive tumor focused radiotherapy group, but the intestinal tract may be included in the target volume (especially PTV_Lar_60&PTV2_Lar_60). If flatus is present in the overlapping area of intestinal tract and target volume, the target dose would be calculated to be low and an unnecessarily high dose might be administered to satisfy the dose constraints. Therefore, it is considered that the intestinal gas in the target volume should be allowed to be overwritten with water-equivalent density. If it would be allowed, please add it to the protocol. 2. High quality of TURBT is needed to accomplish best treatment result. Please mention more about the role of TURBT in the protocol. The reviewer believe that maximal TURBT should be considered for the enrolled patients before radiation therapy.
---

REVIEWER	Neil B. Desai University of Texas Southwestern Medical Center United States of America Boston Scientific clinical trial funding and consulting in prostate cancer regarding rectal spacer hydrogel
REVIEW RETURNED	08-Oct-2020

GENERAL COMMENTS	This is a well thought out, well-written and coordinated international cooperative protocol by experienced investigators in bladder cancer. A track record and preliminary investigation in the experimental interventions is established for this group. Ability to accrue is derived from prior high level/high impact trials. A key question is whether more contemporary techniques for adaptation can be extrapolated to many centers, but this is the focus of the study itself. The treatment questions are important and flexible to accommodate multiple sensitize regimens and fractionation schemes. One question would be whether the presentation of data from the UK that hypofractionation may indeed be a preferred approach would affect whether the investigators see need to enroll full number cohorts at both conventional and hypofractionated regimens or whether to focus on latter, should enrollment be challenged. But this need not be addressed now.
--

REVIEWER	Petra Kroon UMC Utrecht, the Netherlands
REVIEW RETURNED	12-Oct-2020

GENERAL COMMENTS	Dear authors, RAIDER is an interesting study in which the added value of ART/DART is investigated in a multi-centre setting. Hopefully, the study recruitment is going well.
---

VERSION 1 – AUTHOR RESPONSE

Reviewer: 1

Reviewer Name: mccm hulshof

Institution and Country: Amsterdamumc

Please state any competing interests or state 'None declared': no conflicts of interest

Comments to the Author

The manuscript is a protocol of a running randomized phase II study in muscle invasive bladder cancer comparing whole bladder radiation (standard arm), tumor focused radiotherapy and dose escalated adaptive tumor focused radiotherapy. The study will answer a relevant question in bladder sparing treatment and might be able to improve technique and reducing toxicity in BST. The study protocol is too extended and complicated to be described when publishing the results and thus deserves publication in a separate manuscript. As a reviewer of the manuscript it is not my task to comment on the study design itself. My only comment to the authors would be to add under "strength and limitations" the large accepted heterogeneity in treatment and techniques allowed (dose fractionation, chemotherapy schedules and types of drugs, use of fiducials etc) which might bias the toxicity results.

- We thank the reviewer for recognising the importance of publishing the study protocol as a separate manuscript.

- Partially agree. The issue raised regarding the 'heterogeneity in treatment and technique' is recognised. It has been addressed and minimised within the manuscript and protocol as follows;

- o For dose fractionation, in order to manage the inclusion of both fractionation cohorts, the choice of schedule had to be confirmed by each participating site before trial commencement and then had to be used for all patients at that site. The study is separately powered for each fractionation cohort (essentially 2 trials in 1) and analysis will thus establish the safety of dose escalation with each fractionation regimen independently.

The fact that the two fractionation cohorts will be analysed separately for the primary endpoint is explained on p8 para 3 of the manuscript and, supplementary material, appendix 1 main protocol p14.

- o The concurrent chemotherapy regimes were limited to those four regimes with supporting phase III and phase II outcome data. It was advised that each centre should also use the same regime for each participant within the trial where possible (manuscript p 10, para 3).

- o In addition treatment allocation is by minimisation with a random element; balancing factors were centre, neo-adjuvant chemotherapy use and concomitant radiosensitising therapy use. This is described in the manuscript (p8, para 2) and in the supplementary material, appendix 1 main protocol (p29 under treatment allocation).

Given technique adopted is largely centre driven, we believe the criticism of treatment heterogeneity has been reduced as much as possible by the approaches adopted with the trial.

We have also specified within the strengths and limitations that this is a non-comparative trial design (p4, line 12) and in the main text (p21, para 3).

No further changes made to the text.

Reviewer: 2

Reviewer Name: Liam Mannion

Institution and Country: City, University of London + King's college London. UK

Please state any competing interests or state 'None declared': None declared

Comments to the Author

This is a protocol for a phase II multi-centre randomised controlled trial on muscle invasive bladder cancer. The protocol is well written and thought out, providing sufficient methodological detail. The only limitation is acknowledged by the authors, in that a comparative analysis of treatment regimens is not possible due to randomisation.

- We thank the reviewer for their time and favourable comments. No changes to the manuscript indicated.

Reviewer: 3

Reviewer Name: Takahiro Osawa

Institution and Country: Hokkaido University Hospital, Japan

Please state any competing interests or state 'None declared': None

Comments to the Author

The authors submitted a study in which they tried to test within a randomized multi-center phase II trial whether this technique will enable dose escalation with acceptable rates of toxicity. I think this article is interesting and insightful, and particularly well written. However, the major points you should consider are the following:

- We thank the reviewer for their time and favourable comments.

Major

1. Dose calculation is done on CT30 data set for adaptive tumor focused radiotherapy group, but the intestinal tract may be included in the target volume (especially PTV_Lar_60&PTV2_Lar_60). If flatus is present in the overlapping area of intestinal tract and target volume, the target dose would be calculated to be low and an unnecessarily high dose might be administered to satisfy the dose constraints. Therefore, it is considered that the intestinal gas in the target volume should be allowed to be overwritten with water-equivalent density. If it would be allowed, please add it to the protocol.

- The reviewer raises an important point that was considered at the time of protocol development, i.e. how to accurately represent dose to the intestinal tract/bowel at the time of planning particularly for the large plan (PTV_Lar_60&PTV2_Lar_60).

Intestinal gas/bowel in the target volume at the time of planning was explicitly not chosen to be overridden because it incorrectly assumes that plan selection will be optimal; that the bladder has filled by a magnitude such that it adequately pushes bowel out of the target (justifying large plan selection) and so minimising dose to these structures.

We therefore made the decision to accept that that large plan would overestimate actual dose delivered to bowel if plan selection was correct. Underestimating delivered dose to bowel would have been a more critical oversight given dose escalation (p14 para 6).

The medium (and small) plans were accepted to be closer representatives of actual delivered dose to bowel. And is why the medium plan (and not large) had to meet the pre-specified mandatory dose

constraints for dose escalation (p14 para 6).

This approach was also adopted in the preceding published phase I safety work (Hafeez S et al: Prospective Study Delivering Simultaneous Integrated High-dose Tumor Boost (The supporting rationale for this approach is also described in the main manuscript (p14, para 6) and in supplementary material, appendix 2 radiotherapy planning and delivery guidelines (p24, section 4.4 normal tissue constraints).

No further changes made to the text.

2. High quality of TURBT is needed to accomplish best treatment result. Please mention more about the role of TURBT in the protocol. The reviewer believe that maximal TURBT should be considered for the enrolled patients before radiation therapy.

- We have already stated that 'all participants should have a transurethral resection of the bladder tumour (TURBT)' (p10 para 1).
- The role of TURBT prior to radiotherapy is specified in the main text (p10 para 1) under the study treatment section and in supplementary material, appendix 1 main protocol (p23 section 10.1 pre-trial treatment). It is also included in Table 1 schedule of assessments.

The protocol did not mandate that the TURBT had to be visibility complete prior to radiotherapy. This is because it is accepted that inability to perform a complete TURBT does not necessarily preclude patients accessing bladder preservation with radiotherapy although we acknowledge that complete TURBT carries prognostic value.

No further changes made to the text.

Reviewer: 4

Reviewer Name: Neil B. Desai

Institution and Country: University of Texas Southwestern Medical Center

United States of America

Please state any competing interests or state 'None declared': Boston Scientific clinical trial funding and consulting in prostate cancer regarding rectal spacer hydrogel

Comments to the Author

This is a well thought out, well-written and coordinated international cooperative protocol by experienced investigators in bladder cancer. A track record and preliminary investigation in the experimental interventions is established for this group. Ability to accrue is derived from prior high level/high impact trials. A key question is whether more contemporary techniques for adaptation can be extrapolated to many centers, but this is the focus of the study itself. The treatment questions are important and flexible to accommodate multiple sensitize regimens and fractionation schemes. One question would be whether the presentation of data from the UK that hypofractionation may indeed be a preferred approach would affect whether the investigators see need to enroll full number cohorts at both conventional and hypofractionated regimens or whether to focus on latter, should enrollment be challenged. But this need not be addressed now.

- We thank the reviewer for their time, favourable and thoughtful comments. No changes to the manuscript indicated.

Reviewer: 5

Reviewer Name: Petra Kroon

Institution and Country: UMC Utrecht, the Netherlands

Please state any competing interests or state 'None declared': None

Comments to the Author

Dear authors, RAIDER is an interesting study in which the added value of ART/DART is investigated in a multi-centre setting. Hopefully, the study recruitment is going well.

- We thank the reviewer for their time and favourable comments. No changes to the manuscript indicated.